# Use of the Probiotic *Bifidobacterium animalis* subsp. *lactis* HN019 in Oral Diseases

**DOI:** 10.3390/ijms23169334

**Published:** 2022-08-19

**Authors:** Lisa Danielly Curcino Araujo, Flávia Aparecida Chaves Furlaneto, Léa Assed Bezerra da Silva, Yvonne L. Kapila

**Affiliations:** 1Department of Pediatric Dentistry, School of Dentistry of Ribeirão Preto, University of São Paulo, Ribeirão Preto 14049-900, SP, Brazil; 2Department of Oral & Maxillofacial Surgery, and Periodontology, School of Dentistry of Ribeirão Preto, University of São Paulo, Ribeirão Preto 14049-900, SP, Brazil; 3Department of Dentistry, Sections of Biosystems and Function and Periodontics, School of Dentistry, University of California Los Angeles, Los Angeles, CA 90095, USA

**Keywords:** probiotic, endodontics, oral health, dysbiosis, periodontitis

## Abstract

The oral cavity is one of the environments on the human body with the highest concentrations of microorganisms that coexist harmoniously and maintain homeostasis related to oral health. Several local factors can shift the microbiome to a pathogenic state of dysbiosis. Existing treatments for infections caused by changes in the oral cavity aim to control biofilm dysbiosis and restore microbial balance. Studies have used probiotics as treatments for oral diseases, due to their ability to reduce the pathogenicity of the microbiota and immunoinflammatory changes. This review investigates the role of the probiotic *Bifidobacterium animalis* subsp. *lactis* (*B. lactis*) HN019 in oral health, and its mechanism of action in pre-clinical and clinical studies. This probiotic strain is a lactic acid bacterium that is safe for human consumption. It mediates bacterial co-aggregation with pathogens and modulates the immune response. Studies using *B. lactis* HN019 in periodontitis and peri-implant mucositis have shown it to be a potential adjuvant treatment with beneficial microbiological and immunological effects. Studies evaluating its oral effects and mechanism of action show that this probiotic strain has the potential to be used in several dental applications because of its benefit to the host.

## 1. Introduction

Of all the microbiomes in the human body, the gut microbiome contains the greatest number of microbes [1]. In a state of health, the interactions of the microorganisms with epithelial cells, nerve cells, the immune system, and endocrine cells of the intestine mediate the homeostasis of this site [2,3,4]. The healthy gut microbiome has the ability to produce vitamins, absorb nutrients and ferment fibers, in addition to improving immunity and bringing energy to the body, thus demonstrating its fundamental role in the overall health of the individual [5].

The intestine is one of the main targets for the use of probiotics [3,4], defined as live microorganisms that, when administered in adequate amounts, improve the health of the host [6,7]. Thus, probiotics are used as preventive or therapeutic strategies in various conditions associated with disease, such as obesity, pre-diabetes and diabetes, gastrointestinal diseases, functional abdominal pain, atopic dermatitis, and for oral applications) [7,8,9,10,11,12,13,14].

Consumed for more than 1000 years by humans, probiotics are present in fermented products, such as cheeses, breads, yogurts, and non-fermented products, such as meats and fruits, as well as in industrialized supplements. The actions of probiotics are mediated primarily through their ability to inhibit pathogens through the production of bactericidal bioactive peptides, called bacteriocins, lactic acid, hydrogen peroxide, and bacteriocin-like inhibitory substances, or competing with pathogens for nutrients and binding sites, or killing pathogens [15,16,17]. In addition, probiotics improve epithelial barrier function by modulating signaling pathways and the local and systemic host immune-inflammatory response [18,19].

Lactic acid bacteria (LAB) are generally Gram-positive bacteria that are characterized by their ability to produce bacteriocins, alter the pH, and modulate the immunoinflammatory system [18,20,21,22]. LAB can be used as probiotics in clinical settings so long as there is adherence to strict selection guidelines [23]. One of the main actions of LAB is mediated by the effects of their products and by-products inside other bacterial cells [18,24]. Organic acids, such as acetic acid and lactic acid, produced by these bacteria are responsible for their inhibitory activity on pathogens. These acids enter bacterial cells and dissociate within the cytoplasm, leading to a decrease in pH and intracellular accumulation, which, in turn, kills the pathogenic bacteria. Another product of LAB are bacteriocins, which act against closely related bacteria and can lead to the destruction of target cells via pore formation and/or cell wall inhibition [18,25,26].

*Bifidobacterium animalis* subsp. *lactis* (*B. lactis*) HN019 is a LAB that was isolated from yogurt. Its complete genomic sequence was published in 2018, which allowed for strict control, quality, safety and purity of the *B. lactis* HN019 strain [3,27,28]. *B. lactis* HN019 has several gut health benefits, such as improving intestinal mobility and relieving constipation, providing defense against intestinal pathogens, and improving macronutrient absorption [3,29]. It also has benefits in regulating the immune response, such as modulating inflammatory and oxidative biomarkers in healthy subjects and also in patients with systemic diseases, including metabolic syndrome [30,31].

The therapeutic use of probiotics in healthcare and in dentistry has grown significantly in recent years [4,32,33,34,35,36]. The oral cavity contains the second largest microbiome in the human body [37,38,39]. It is colonized by a great diversity of microorganisms, which interact dynamically with each other or with other microbiota from other sites, such as the microbiota of the gut [39]. In a single day, an individual can swallow approximately 600 mL of saliva containing microorganisms from the mouth, which reach the intestine through the gastrointestinal tract [40]. Thus, it is believed that pathogenic bacteria originating in the oral cavity as a result of poor hygiene or alteration of the oral microbiota, will promote a disturbance in the intestinal microbiota, which may lead to inflammation and systemic changes. This highlights the importance of having a balanced and homeostatic microbiota that promotes the health of the patient [41,42]

From the 2000s, studies started emerging on the preventive and therapeutic effects of probiotics in the context of the oral cavity for the management of caries, periodontal diseases, halitosis and periapical lesions [35,43,44,45]. However, a focused review of the role of the probiotic *B. lactis* HN019 in oral health has not been presented. Thus, the objective of this literature review is to describe the mechanism of action and the effects of the probiotic *B. lactis* HN019 in the treatment of different oral diseases, drawing from preclinical and clinical studies.

## 2. Mechanism of Action of *Bifidobacterium animalis* subsp. *lactis* (*B. lactis*) HN019

There is a current focus on trying to understand the intricate mechanisms of action of bacteria on the host, for the purposes of discovering new therapeutic approaches and new materials for promoting health [4,27,46]. Although the mechanism of action of probiotics in oral health is not yet fully understood, it is known that an imbalance in the composition of the microbiota, known as dysbiosis, can cause changes to and negatively impact the oral cavity [47]; whereby probiotics might offer a solution. This dysbiosis can be caused by a loss of beneficial microorganisms and an increase in pathogens, as well as by the loss of microbial diversity [20,47]. In addition, this dysbiosis is associated with increased permeability and disruption of the epithelial barrier, leading to inflammation and chronic inflammatory pathologies [48]. However, it is not known whether the dysbiosis is a cause or a consequence of this change [20] (Figure 1).

The strain *B. lactis* HN019 has been documented to be part of human food consumption since 1980, but it is believed that it has been used for this application even before this date [49]. The safe and effective daily dose of this strain ranges from 10^7^ to 10^11^ colony forming units (CFU) per day, and it can be consumed for at least 7 days and for up to 2 years [4]. It has an excellent capacity to adhere to epithelial cells, a high tolerance for and ability to survive in a low pH environment, resistance to bile salts [4,27,50,51], and ability to modulate the immune response [4,27].

Studies show that this strain acts on intercellular junctions, especially tight junctions [4,52]. Tight junctions are located in the most apical region of the cell, and are comprised of two proteins (claudin and occludin), which are responsible for establishing the epithelial barrier that prevents entry of macromolecules (lipids and proteins) [20,47,53]. These junctions are essential for regulating the permeability of the epithelium, which when altered promotes inflammation and, consequently, the development of disease [20,53]. *B. lactis* HN019 can increase the strength of these junctions by modulating transepithelial electrical resistance, thereby preventing increases in epithelial cell permeability, although these changes were not statistically significant in a cell-free supernatant assay [4,52].

There are multiple mechanisms that are involved in mediating probiotic effects in the gut-bone axis. Modifications to the epithelial barrier in the gastrointestinal tract may control bone health [36]. In addition to modulating and promoting intestinal health, hormones and immune cells present in the gastrointestinal tract act to maintain bone health by regulating the balance between resorption by osteoclasts and bone neoformation by osteoblasts. The epithelial barrier plays a key role in the absorption of components involved in bone mineralization, such as calcium, phosphorus and magnesium, and they produce endocrine cell factors that carry signals to bone cells, such as incretin and serotonin [14,20]. Therefore, due to these different mechanisms, the manipulation of the microbiota can assist in bone development [36]. The receptor activator of NF-kappaβ (RANK), receptor activator of NF-kappaβ ligand (RANKL) and osteoprotegerin (OPG) system responsible for osteoclastogenesis can be regulated by modifications in the microbiota [14]. These potential mechanisms may also be at play in the context of bone changes in periodontal disease, since *B. lactis* HN019 can decrease several bone loss parameters in experimental periodontitis in rats [54].

The mouth harbors a diverse and distinct microbiome, as a result of different hard non-shedding surfaces and shedding cellular surfaces [38,39]. Because the oral cavity interacts significantly with the external environment, its microbiome diversity is of extreme importance for the maintenance of both oral and systemic health [39]. The process of colonization of the oral cavity is dynamic, starting at birth and it continues until the completion of the permanent dentition [55,56]. When factors such as poor oral hygiene, diet and immunodeficiency alter the existing microbial balance of the microbiome, infectious diseases of a polymicrobial nature set in [35,57,58]. Microorganisms and toxins that destroy tissues are present together and organized into biofilms, which produce substances known as extracellular polysaccharides that assist in microbial aggregation and co-aggregation and adhesion to surfaces of the oral cavity: enamel, cementum and dental implant surfaces [35,59,60,61,62,63]. These biofilms can change depending on the disease entity and depending on their location, such that each altered biofilm is associated with each disease entity, including caries, periodontitis and endodontic infections [35,63]. In this context, *B. lactis* HN019 is currently being studied for the purpose of altering the dysbiosis of biofilms present in oral diseases, with beneficial results noted in periodontitis and peri-implant mucositis.

## 3. *B. lactis* HN019 and Periodontitis

Chronic periodontitis is a polymicrobial disease that is highly prevalent and is characterized by an inflammatory process that affects the supporting tissues of the teeth, but is also associated with other systemic changes/diseases when left untreated [64,65]. Conventional treatment for periodontal disease is often not effective in controlling inflammation, and many patients experience disease recurrence [64,65,66,67].

Cases of recurrent periodontitis are treated with adjuvant antibiotic therapy. However, the possibility of bacterial resistance has been considered a problem after the excessive use of these drugs, leading to the search for new agents to control infectious diseases [64,65,68].The development of therapeutic alternatives that can act as adjuvants to clinical treatment is essential;, thus, probiotics have emerged and have shown satisfactory and effective results [64,67,68,69,70].

Although probiotics are not yet used as alternative treatments in clinical practice, interest in their use for the treatment of periodontal diseases has grown worldwide [71]. *Lactobacillus* and *Bifidobacterium* strains have been highlighted for their ability to alter periodontal biofilms and modulate the immunoinflammatory response [70,71,72].Studies testing *Bifidobacterium animalis* found decreased biofilm virulence, decreased gingival inflammation, and decreased levels of pathogenic bacteria relevant to periodontal diseases [73,74,75,76].

Ricoldi et al. (2017) evaluated the oral administration of the probiotic *B. lactis* HN019 as an adjunctive treatment to scaling and root planing (SRP) in rats with experimental periodontitis. The animals treated with *B. lactis* HN019 and SRP showed reduced attachment loss and alveolar bone resorption, when compared with those treated with SRP alone. There were also lower numbers of osteoclasts and pro-inflammatory cytokines, and higher levels of anti-inflammatory cytokines in the probiotic group [68].In addition to being used as an adjuvant treatment, an increasing number of studies have demonstrated the use of subgingival probiotics, that is, as topical treatment in periodontal pockets [77]. Oliveira et al. (2017) evaluated the effects of topical administration of *B. lactis* HN019 in rats with experimental periodontitis. Histological and micro computed tomography analyses demonstrated that topical administration of this probiotic strain decreased bone resorption, thereby demonstrating a protective effect on the alveolar bone. In addition, there was modulation of the immunoinflammatory response and microbiological profile as assessed by cytokine analyses and microbiological assessment [57]. A study by Oliveira et al. (2022) demonstrated a useful vehicle for the delivery of *B. lactis* HN019, by showing that this probiotic delivered with milk and in the absence of mechanical treatment potentiated the effects of the probiotic therapy on periodontal lesions [78].

Other preclinical studies using the ligature-induced periodontitis model demonstrated the potential of *B. lactis* HN019 on systemic diseases associated with periodontitis [79]. Silva et al. (2022) evaluated the effects of this strain on the development of periodontitis associated with metabolic syndrome. The groups with experimental periodontitis +/− metabolic syndrome treated with the probiotic exhibited lower levels of alveolar bone loss compared to the groups not treated with the probiotic. Higher levels of interleukin (IL)-1β and RANKL/OPG were observed in the group with metabolic syndrome that did not receive the probiotic compared to the group which received the probiotic. Furthermore, the serum levels of total cholesterol and triglycerides of the animals with metabolic syndrome and treated with the probiotic were statistically lower than those not treated with the probiotic. Cardoso et al. (2020) analyzed the effects of systemic administration of *B. lactis* HN019 on ligature-induced periodontitis in rats with experimental rheumatoid arthritis. They showed that the probiotic group exhibited reduced bone loss, tumor necrosis factor-α and IL-6 levels, and increased IL-17 levels. Therefore, these studies demonstrated that *B. lactis* HN019 modulated immunoinflammation and systemic parameters, reducing the severity of periodontitis associated with systemic diseases [80,81].

Invernici et al. (2018), in a randomized clinical study, revealed that patients with generalized chronic periodontitis who used lozenges containing the probiotic *B. lactis* HN019 had better clinical results, with a 58% reduction in deep periodontal pockets in contrast with a 22% reduction in patients who used the placebo [82]. The microbiological analysis revealed that only the probiotic group exhibited a lower concentration of pathogenic microorganisms, in addition to having a lower concentration of pro-inflammatory cytokines when compared to the control group. Furthermore, Invernici et al. (2020) showed that improvements in the clinical parameters (gingival bleeding and plaque accumulation) were superior in the group of periodontitis patients that received *B. lactis* HN019 lozenges compared to the placebo group. An increase in the immunocompetence of the oral epithelial barrier was demonstrated in the patients treated with the probiotic, with increased expression of beta-defensin (BD)-3, toll-like receptor 4 (TLR4), and cluster of differentiation (CD)-4 [83]. Table 1 summarizes the vehicles, dosages and results of studies of this probiotic strain in periodontitis.

In the context of periodontitis, *B. lactis* HN019 is being studied intensively, showing effective and promising results [54,68,78,79,80,81,82,83], and requiring further clinical studies to demonstrate its full clinical utility for routine care (Figure 2).

The effects of probiotics on bone led to the investigation of their use in orthodontic tooth movement. Duffles et al. (2022) investigated the impact of *B. lactis* HN019 on bone remodeling induced by orthodontic tooth movement in mice. Although microtomography analysis showed that probiotic treatment did not modify the alveolar bone, the therapy did restrain the tooth movement. Groups treated with the probiotic had a decreased number of osteoclast cells and a higher concentration of short-chain fatty acid in their feces [84].

## 4. *B. lactis* HN019 and Peri-Implant Mucositis

Currently, the use of dental implants in oral rehabilitation is increasing in popularity, because of their ability to address aesthetic and functional needs and due to their durability and high success rates [85]. However, there are frequent complications with dental implants, such as peri-implant mucositis and peri-implantitis [86,87]. Healthy peri-implant mucosa is composed of keratinized epithelium and its basal lamina faces the surface of the dental implant or abutment [86,88]. When there is an accumulation of biofilm at the abutment-mucosal interface, an inflammatory lesion may develop. However, it is a reversible change, without radiographic bone loss, thus differentiating it from peri-implantitis [86,89].

Conventional non-surgical mechanical therapy is still a standard treatment used to treat peri-implant diseases [90]. However, since this treatment does not completely eliminate the bacterial biofilm, several biological and chemical agents and methods have been used as alternative or adjuvant treatments, such as antibiotics, laser therapy, chlorhexidine, enamel matrix derivatives, and probiotics [90,91].

Several probiotic strains have been used to treat peri-implant diseases [91]. In a randomized clinical trial, *B. lactis* HN019 was tested along with two other probiotic strains (*L. rhamnosus* HN001 and *L. paracasei* Lpc-37) as adjuvants to mechanical debridement in patients presenting with peri-implant mucositis. The probiotics were administered topically as well as ingested by the patients. After 24 weeks, the group using the probiotics showed no bleeding in 72.2% of the patients, demonstrating that there was a better clinical effect with the probiotics than the control group, which received debridement alone. In addition to the clinical benefits, a lower concentration of pro-inflammatory cytokines was also observed [92]. However, this was the first study using this probiotic strain in a dental implant setting. Thus, further preclinical and clinical studies are needed to validate these results and to confirm the beneficial role of *B. lactis* HN019 in addressing peri-implant mucositis.

## 5. Conclusions

The probiotic strain *B.lactis* HN019 has been shown to play a role in modulating the immunoinflammatory response of the human organism, including promoting oral health benefits, when administered in adequate doses. Furthermore, the use of this strain in periodontitis is supported by data from pre-clinical studies and clinical trials. However, due to the significant potential of this strain to maintain oral health, further studies on several oral diseases are needed, to expand the current findings and to determine the application of *B. lactis* HN019 in other oral diseases, like peri-implant mucositis.

## Figures and Tables

**Figure 1 ijms-23-09334-f001:**
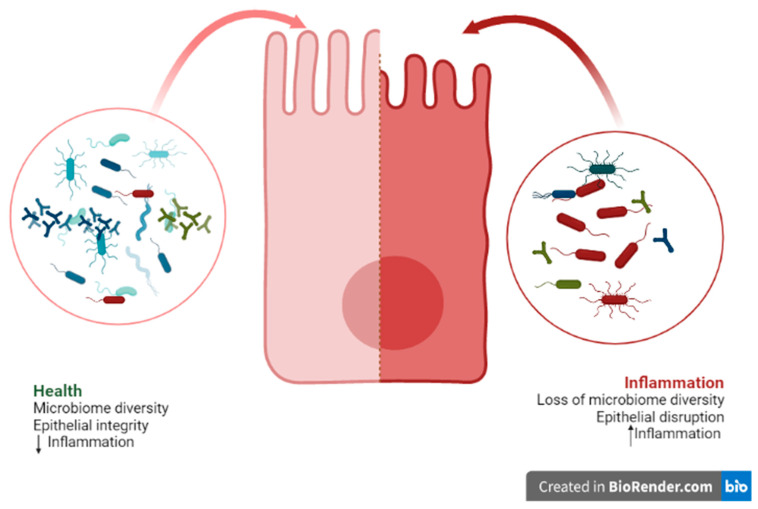
Representative figure of an epithelial cell and its interactions with the microbiome.

**Figure 2 ijms-23-09334-f002:**
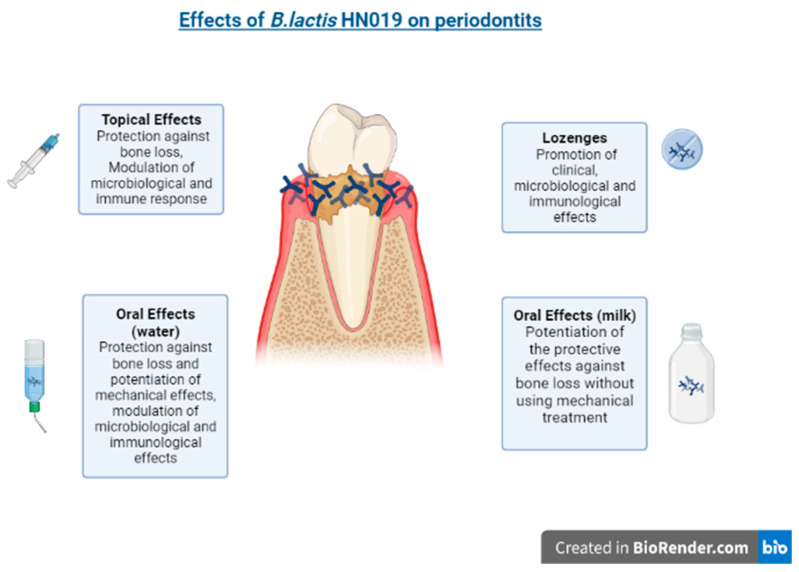
The effects of *B. lactis* HN19 on periodontitis.

**Table 1 ijms-23-09334-t001:** Studies using the *B. lactis* HN019 strain in the context of periodontitis.

Administration	Type of Study	Dosage/Time	Result	Reference
10% skimmed milk (Molico^®^, Nestle Brasil Ltd.a., São Paulo, SP, Brazil) plus *B. lactis* HN019	Preclinical in rats	1.9 × 10^9^; 15 days	Oral use of *B. lactis* HN019 potentiated the effect of SRP	Ricoldi et al., 2017 [68]
Suspension containing *B. lactis* HN019 plus 2% carboxymethylcellulose, applied topically via irrigation	Preclinical in rats	1.9 × 10^9^; 0, 3 and 7 days	Topical use of *B. lactis* HN019 promoted a protective effect against alveolar bone loss and modified the immunoinflammatory and microbiological response	Oliveira et al., 2017 [54]
*B. lactis* HN019 with water and *B. lactis* HN019 with milk	Preclinical in rats	1.9 × 10^9^; 4 weeks	*B. lactis* delivered with milk, and without mechanical treatment, potentiated the protective effects of HN019 in rats with experimental periodontitis	Oliveira et al., 2022-a [78]
*B. lactis* HN019 with water	Preclinical in rat	1 × 10^9^; 4 weeks	*B. lactis* HN019 administration before and during periodontitis development in rats promotes a protective effect against alveolar bone loss by modifying local and systemic microbiologicaland immunoinflammatory parameters.	Oliveira et al., 2022-b [79]
B. *lactis* HN019 suspended in deionized water for the treatment of induced periodontitis and rheumatoid arthritis	Preclinical in rats	1.5 × 10^9^; 39 days	Systemic administration of *B. lactis* HN019 promoted a protective effectagainst the destruction of periodontal tissue, decreasing both bone loss and inflammatory mediators and increasing theproportion of bacteria compatible with periodontal health in rats with experimental rheumatoid arthritis and periodontitis	Cardoso et al., 2020 [80]
*B. lactis* HN019 with water	Preclinical in rats	1 × 10^9^; 8 weeks	*B. lactis* HN019 reduced the severity of periodontitis inrats with metabolic syndrome, modulating both systemic and immunoinflammatory parameters. The probiotic led to reduction in total cholesterol and triglycerides levels.	Silva et al., 2022 [81]
*B. lactis* HN019 in lozenges	Randomized clinical trial	10^9^; 30 days	Ancillary treatment with *B. lactis* HN019 promoted positive clinical, microbiological and immunological effects	Invernici et al., 2018 [82]
*B. lactis* HN019 in lozenges	Randomized clinical trial and in vitro analysis	10^9^; 30 days	*B. lactis* HN019 showed immunological and antimicrobial properties	Invernici et al., 2020 [83]

## Data Availability

Not applicable.

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
