# Peer review of "Use of the Probiotic Bifidobacterium animalis subsp. lactis HN019 in Oral Diseases"

_ijms, 2022, doi:10.3390/ijms23169334_

Round 1

Reviewer 1 Report

The authors aimed to describe the mechanism of action and the effects of the probiotic B. lactis HN019 in the treatment of different oral diseases, drawing from preclinical and clinical studies.

The study covers some issues that have been overlooked in other similar topics. The structure of the manuscript appears adequate and well divided in the sections. Moreover, the study is easy to follow, but some issues should be improved. Some of the comments that would improve the overall quality of the study are:

a. Authors must pay attention to the technical terms acronyms they used in the text.

b. English language needs to be revised.

c. Conclusion Section: This paragraph required a general revision to eliminate redundant sentences and to add some "take-home message".

Author Response

Reviewer 1

The authors aimed to describe the mechanism of action and the effects of the probiotic B. lactis HN019 in the treatment of different oral diseases, drawing from preclinical and clinical studies.

The study covers some issues that have been overlooked in other similar topics. The structure of the manuscript appears adequate and well divided in the sections. Moreover, the study is easy to follow, but some issues should be improved. Some of the comments that would improve the overall quality of the study are:

Question a: Authors must pay attention to the technical terms acronyms they used in the text.

Answer a: Thank you very much for your suggestion. We have checked and revised the entire text, with special attention to the acronyms.

Question b: English language needs to be revised.

Answer b: Thank you very much for your comment. We revised the text accordingly.

Question c: Conclusion Section: This paragraph required a general revision to eliminate redundant sentences and to add some "take-home message".

Answer c: Thank you for your suggestion. We rewrote the conclusion as follows:

Previously: “The studies published thus far highlight the potential benefits of using B. lactis HN019 in an oral setting. The data indicate that the use of this probiotic strain can restore the health of the oral cavity, and consequently promote the health of the host. The use of B. lactis HN019 in a periodontitis setting is supported by data from pre-clinical studies and clinical trials. Further studies are needed to expand the current findings and to determine the application of B. lactis HN019 in other oral diseases, like peri-implant mucositis.”

Now: “The probiotic strain B. lactis HN019 has been shown to play a role in modulating the immunoinflammatory response of the human organism, including promoting oral health benefits, when administered in adequate doses. Furthermore, the use of this strain in periodontitis is supported by data from pre-clinical studies and clinical trials. However, due to the significant potential of this strain to maintain oral health, further studies on several oral diseases are needed, to expand the current findings and to determine the application of B. lactis HN019 in other oral diseases, like peri-implant mucositis.”

Reviewer 2 Report

This review is well mentioned about the studies published thus far highlight the potential benefits of using B. lactis in an oral setting. The data indicate that the use of this probiotic strain can restore the health of the oral cavity, and consequently promote the health of the host. The use of B. lactis HN019 in a periodontitis setting is supported by data from pre-clinical studies  and clinical trials. 

Author Response

Reviewer 2

This review is well mentioned about the studies published thus far highlight the potential benefits of using B. lactis in an oral setting. The data indicate that the use of this probiotic strain can restore the health of the oral cavity, and consequently promote the health of the host. The use of B. lactis HN019 in a periodontitis setting is supported by data from pre-clinical studies and clinical trials.

 Answer to reviewer 2: Thank you so much for your comment. We are grateful that you understood the aim and purpose of this review and we hope that it will add to the scientific community.